# Tailoring Visual Object Representations to Human Requirements: A Case Study with a Recycling Robot

**Debasmita Ghose, Michal Adam Lewkowicz, Kaleb Gezahegn, Julian Lee,**\*
**Timothy Adamson,**\* **Marynel Vázquez, Brian Scassellati**
Yale University
{first name}.{last name}@yale.edu

**Abstract:** Robots are well-suited to alleviate the burden of repetitive and tedious manipulation tasks. In many applications, though, a robot may be asked to interact with a wide variety of objects, making it hard or even impossible to pre-program visual object classifiers suitable for the task of interest. In this work, we study the problem of learning a classifier for visual objects based on a few examples provided by humans. We frame this problem from the perspective of learning a suitable visual object representation that allows us to distinguish the desired object category from others. Our proposed approach integrates human supervision into the representation learning process by combining contrastive learning with an additional loss function that brings the representations of human examples close to each other in the latent space. Our experiments show that our proposed method performs better than self-supervised and fully supervised learning methods in offline evaluations and can also be used in real-time by a robot in a simplified recycling domain, where recycling streams contain a variety of objects.

**Keywords:** Contrastive Representation Learning, Human Centric Robot Learning

## 1   Introduction

Robots are well-suited to alleviate the burden of repetitive and tedious manipulation tasks in homes and in the public and private sectors. For example, robots might help humans with housekeeping, janitorial and custodial work, or sorting recyclables. In many of these applications, a robot may be asked to interact with a wide variety of objects, making it hard or even impossible to pre-program visual object classifiers suitable for the task of interest. For example, imagine a home robot meant to help a person clean her room. One might think of defining object categories for this task, such as "clothes", "shoes", "books", etc. But what if the human wants the robot to pick up only "folded clothes" from the floor? These kinds of tasks require learning from the human what object category is relevant. Since these categories can be defined arbitrarily, the robot needs to learn the object properties that distinguish the human-selected objects from others. Importantly, another challenge in learning an object category from a human is that the robot cannot assume the objects not selected by the human do not belong in the category of interest. That is because humans are unwilling to give many examples and may only provide examples of objects they are interested in.

In this work, we study the problem of learning a classifier for visual objects based on (a few) examples provided by humans. We frame this problem from the perspective of learning a suitable visual object representation that allows the robot to distinguish the desired object category from others. This perspective is motivated by recent work in self-supervised learning, which has led to significant advances in learning useful representations from high-dimensional data [1]. In particular, contrastive learning [2] learns representations by pulling similar samples close to each other while pushing dissimilar ones far apart in the latent space. Recent contrastive learning methods have been successful in learning strong representations for various downstream tasks in computer vision, such as image classification, semantic segmentation, and object detection [1]. The challenge with applying existing contrastive learning techniques to the problem of learning a representation that distinguishes an arbitrary object category based on human examples is that contrastive learning

---

\*Authors contributed equally
   Code and data available at https://sites.google.com/view/corl22-contrastive-recycling/home

6th Conference on Robot Learning (CoRL 2022), Auckland, New Zealand.

will capture properties of the data, but those properties may not necessarily lead to a good object classifier according to human requirements.

We propose a new approach that integrates human supervision into the representation learning process to compute visual object representations in accordance with human requirements for the task of interest. Our approach combines an existing contrastive learning method that maximizes the agreement between different instances in the same cluster with our proposed loss function that brings the representations of objects selected by a human close to each other in the latent space. This combination results in representations that encode the human desired object characteristics.

We evaluate our approach to incorporate human supervision into the representation learning process in a simplified recycling setup derived from a standard Material Recovery Facility (MRF). We focus on this challenging application domain for three reasons. First, recycling streams can contain a wide variety of diverse objects, ranging in size, weight, color, cleanliness, and form [3]. Second, constraints on what needs to be recycled can vary dynamically because the requirements of the MRF's customers may vary [4]. Third, because of the many objects that quickly move by, a human demonstrator cannot pick out all the items that need to be removed from the stream. This results in limited positive examples and no negative examples for the robot.

To conduct offline and online evaluations, we set up a conveyor belt that humans and robots can work on simultaneously and a feeder system to provide a steady stream of recyclables. We demonstrate that our method outperforms a fully-supervised learning method and some self-supervised learning techniques on an offline dataset. Finally, we deploy our proposed learning method in real-time on a robot working alongside a human in the simplified recycling setup. Our results show how incremental supervision from the human helps the robot learn visual object representations tailored to the human's requirements.

## 2 Related Work

**Self-Supervised Learning:** Self-supervised learning obtains supervisory signals from the data by leveraging its underlying structure [1, 5]. One broad category of techniques in self-supervised learning is contrastive learning, which has been used in a wide range of computer vision applications [6, 7, 8]. Contrastive learning methods [2] typically learn a latent space by pushing similar samples (e.g., two adjacent frames of a video [9]) close to each other while pulling dissimilar data samples (e.g., frames from two different videos) apart in the latent space. More recently, *instance discrimination* [10, 11, 12, 13], a variant of contrastive learning, has shown remarkable success in improving performance on a variety of downstream tasks. To learn latent-space representations, most instance discrimination techniques use two augmented views of the same image as a positive data pair in conjunction with a variety of contrastive learning techniques, such as the contrastive loss [10, 11, 14], triplet loss [15, 16], momentum encoding [17, 18, 19], or online clustering [20].

Recently, there have been some efforts to learn more robust latent-space representations by leveraging properties of non-trivial positive pairs that are correlated in the latent space. For example, these non-trivial positive pairs can either be selected from a support set [12, 21] or from associating multiple data instances with clusters [6, 13, 22, 23]. In this work, we leverage a clustering technique called Contrastive Clustering [13] to enable robots to learn visual object representations. Contrastive Clustering extends instance-level discrimination to distinguish between objects belonging to different clusters while maximizing agreement between objects belonging to a cluster.

**Adding Supervision to Contrastive Learning:** Prior work has guided representation learning processes by leveraging labeled data [24, 25]. Khosla et al. [25] proposed a method to train a contrastive learner in a fully-supervised fashion to improve performance on multiple downstream tasks. However, since their method is label intensive, more recent work has explored using semi-supervised [26, 27] and weakly supervised [28] learning techniques to compute strong representations. Wilber et al. [29] introduced an algorithm called SNaCK that proposes adding a loss function to visual representation learning that reflects a human's preferences. However, SNaCK requires a human to provide a large number of examples of objects that the human thinks are similar and dissimilar, which would be infeasible in real human-robot interactions. To the best of our knowledge, our method is the first to show how human feedback can be leveraged to learn visual object representations tailored to dynamic human-defined requirements with limited examples of a single category.

**Human Centric Robot Learning:** There are many ways in which a robot can learn from a human. The three most common types of demonstrations used for robot learning are kinesthetic teaching,

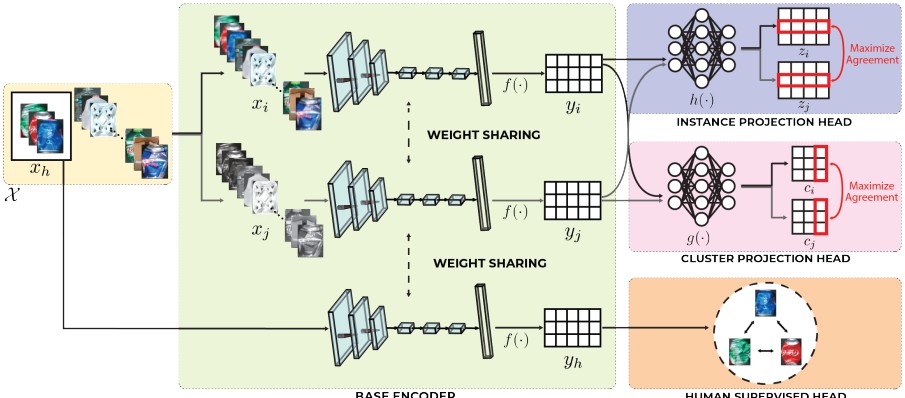

Figure 1: **Our Approach**. Data pairs are constructed using randomly applied data augmentations. The extracted features are fed into three jointly trained heads. The Instance Projection Head projects the features into a space where each row denotes an augmentation of an image and uses the instance loss to minimize the distance between two augmentations of the same image. The Cluster Projection Head projects feature vectors into a space where each column denotes each instance's cluster assignments, and the cluster loss minimizes the distance between instances belonging to a cluster. Finally, the Human-Supervised Head minimizes the distance between each human-selected example.

teleoperation, and passive observation [30]. Our work uses passive observations because of the ease at which they allow the human to teach the robot and the hardware-agnostic nature of this demonstration type. Much of the recent work in the field of learning from passive observations uses reinforcement learning to train a robot. For example, the work of Mukherjee et al. [31] uses a learned goal proximity function as a dense reward for policy training, and Karnan et al. [32] propose to learn navigation policies from a single video demonstration. Our work complements these efforts by proposing a novel approach for learning visual object representations that a robot can use for manipulation tasks, like sorting recycling streams.

Typically within the area of human preference learning, a robot learns the preferences of its human collaborators to perform a task through various reward shaping techniques [33, 34]. Our work is related to this thread of research because we want a robot to learn the preferences of its human collaborator for a sorting task. However, we want the robot to use passive observation rather than more traditional active querying techniques [35, 36, 37] because, in the recycling domain, humans are already busy picking objects from recycling streams. Preference learning in our work then entails having a robot learn the visual characteristics of the categories of objects that the human collaborator is interested in. We propose to approach this problem from the perspective of self-supervised representation learning and incorporate human supervision in a novel manner to ensure that the learned representation aligns well with the human's preferences.

## 3   Learning Visual Object Representations Tailored to Human Requirements

We propose a method to guide the representation learning process of a contrastive learner using human supervision, as shown in Fig. 1. To achieve this goal, we optimize a feature extractor jointly with three "heads." These heads maximize the agreement between 1) multiple views of the same image, 2) multiple instances of the same cluster, and 3) human-selected examples.

Let $\mathcal{X}$ be the set of all objects relevant to a manipulation task and $\mathcal{C}$ be the total number of human-defined categories such that a given category $c \in \{1,..., \mathcal{C}\}$. As a property of our problem, a human selects a set of items $\mathcal{H}$ such that $\mathcal{H} \subset \mathcal{X}$ and all objects in $\mathcal{H}$ belong to a given human-defined category $c$. The paragraphs below use this notation to describe the components of our approach.

**Base Encoder:**  The base encoder is a feature extraction network that obtains representations from two stochastically augmented views of the same image [10]. For a given image $\boldsymbol{x}$, the base encoder outputs a feature vector $\boldsymbol{y}$, such that $\boldsymbol{y} = f(\boldsymbol{x})$. Therefore, if two stochastic augmentations of an image are denoted by $\boldsymbol{x}_i$ and $\boldsymbol{x}_j$, their feature representations can be denoted by $\boldsymbol{y}_i = f(\boldsymbol{x}_i)$ and $\boldsymbol{y}_j = f(\boldsymbol{x}_j)$. Additionally, if $\boldsymbol{x}_h$ is an image in $\mathcal{H}$, then its feature representation is $\boldsymbol{y}_h = f(\boldsymbol{x}_h)$. We feed the image representations output by the base encoder to the three heads described next.

**1. Instance Projection Head:**  The Instance Projection Head is a Multi-Layer Perceptron (MLP) with one hidden layer to map the features of two augmented views of the same image to a latent space

[10]. For example, for the representation of an image's first augmentation $\boldsymbol{y}_i$, its representation after passing through the Instance Projection Head is $\boldsymbol{z}_i = h(\boldsymbol{y}_i)$ (Similarly, for the second augmentation, $\boldsymbol{z}_j = h(\boldsymbol{y}_j)$). As in Chen et al. [10], we apply the NT-Xent loss on the representations from the Instance Projection Head. Therefore, for a positive pair of examples $(i, j)$ constructed from two augmented views of the same image, the instance loss w.r.t. the first augmentation is formulated as:

$$l_{ins}^{i,j} = -\log \frac{\exp\left(sim\left(\mathbf{z}_i, \mathbf{z}_j\right)/\tau_{ins}\right)}{\sum_{k=1}^{N} \mathbb{1}_{[k \neq i]} \exp\left(sim\left(\mathbf{z}_i, \mathbf{z}_k\right)/\tau_{ins}\right)} \tag{1}$$

where N is the number of all positive pairs in a mini-batch, $sim(\boldsymbol{u}, \boldsymbol{v}) = \frac{\boldsymbol{u}^T \boldsymbol{v}}{|\boldsymbol{u}||\boldsymbol{v}|}$ is the cosine similarity between the feature vectors, and $\mathbb{1}[k \neq i] \in \{0, 1\}$, which evaluates to 1 iff $k \neq i$. We use $\mathcal{L}_{ins} = \frac{1}{2N} \sum_{i=1}^{N}(l_{ins}^{i,j} + l_{ins}^{j,i})$ to compute the instance loss over all positive pairs in a mini-batch.

**2. Cluster Projection Head:** The Cluster Projection Head [13] learns representations by maximizing agreement between two representations (under different augmentations) belonging to a given cluster. This head is an MLP with one hidden layer followed by a softmax operation. It projects a data instance into a latent space whose dimensionality equals the total number of pre-defined clusters ($\mathcal{K}$), which should be approximately equal to $\mathcal{C}$. Intuitively, the cluster projection head tries to partition the embedding space into a pre-specified number of clusters based on the inter-instance similarity between object features. For an encoder representation of the first augmentation of an image $\boldsymbol{y}_i$, the latent representation after passing through the cluster projection head is $\boldsymbol{c}_i = g(\boldsymbol{y}_i)$ (Similarly, for a second augmentation, $\boldsymbol{c}_j = g(\boldsymbol{y}_j)$). Let $\mathbb{C}_i \in \mathbb{R}^{N \times K}$ be the output of the Cluster Projection Head for a mini-batch of $N$ images under the first augmentation $i$. Then, the $n^{th}$ element in the output $\mathbb{C}_i$ can be interpreted as its probability of belonging to the $k^{th}$ cluster, where $k \in [1, \mathcal{K}]$ can be denoted by $\mathbb{C}_i^{n,k}$. Thus, $\tilde{\boldsymbol{c}}_i \in \mathbb{R}^N$ can be interpreted as the column vector of $\mathbb{C}_i$ under the first augmentation. The cluster loss can be formulated as:

$$l_{clu}^{i,j} = -\log \frac{\exp\left(sim\left(\tilde{\mathbf{c}}_i, \tilde{\mathbf{c}}_j\right)/\tau_{clu}\right)}{\sum_{k=1}^{K} \mathbb{1}_{[k \neq i]} \exp\left(sim\left(\tilde{\mathbf{c}}_i, \tilde{\mathbf{c}}_k\right)/\tau_{clu}\right)} \tag{2}$$

where $\tau_{clu}$ is the cluster-level temperature parameter. Finally, the cluster loss computed by traversing all $\mathcal{K}$ clusters is $\mathcal{L}_{clu} = \frac{1}{2K} \sum_{k=1}^{K}(l_{clu}^{i,j} + l_{clu}^{j,i}) - H(Y)$, where $H(Y) = \sum_{k=1}^{K}(P(\tilde{\boldsymbol{c}}_i)logP(\tilde{\boldsymbol{c}}_i) + P(\tilde{\boldsymbol{c}}_j)logP(\tilde{\boldsymbol{c}}_j))$ is the entropy of cluster assignment probabilities and $P(\tilde{\boldsymbol{c}}_i) = \sum_{n=1}^{N} \frac{\mathbb{C}_i^n}{||\mathbb{C}||}$. The entropy term helps avoid the trivial solution where most instances get assigned to the same cluster as shown by [13, 23].

**3. Human-Supervised Head:** Our proposed human-supervised head aims to guide the representation learning process towards human requirements. Intuitively, it tries to force the representations of the objects selected by the human close to each other in the latent space to serve as a mechanism to help inform the formation of clusters according to the properties of objects important to the human. This loss is applied only to the representations of objects selected by the human $\boldsymbol{y}_h$. Formally, let $\mathbb{Y}_h \in \mathbb{R}^{\mathcal{H} \times \mathcal{D}}$ be the representation learned by the base encoder across all objects selected by the human ($\mathcal{H} \subset \mathcal{X}$) and $\mathcal{D}$ be the latent dimension of the base encoder. Additionally, let $\mu_h \in \mathbb{R}^{\mathcal{D}}$ be the mean of the representations of all the objects selected by the human, which is considered the cluster-centroid of the human pool.

As new objects are selected by the human, we pass their image ($\boldsymbol{x}_h$) through the base encoder to get their feature representation ($\boldsymbol{y}_h = f(\boldsymbol{x}_h)$). If previously there were $s$ human-selected objects and the human selects $t$ new objects, we re-calculate the mean of the cluster ($\mu_h'$) to include the features of the new objects by $\mu_h' = \frac{\mu_h * s + \sum_t \boldsymbol{y}_t}{s+t}$. The human-supervised loss then minimizes the distance between the new centroid of the human-selected objects and every object selected by the human:

$$\mathcal{L}_{human} = -dist(\mu_h', \mathbb{Y}_h) \tag{3}$$

Theoretically, any distance metric ($dist$) can be used to maximize agreement between the centroid of the human-selected pool and all human-selected objects. However, we empirically found cosine-similarity to work better compared to L1 and L2 distances since it is bounded between -1 and 1.

**Objective Function**: The Instance Projection Head, Cluster Projection Head, and Human Supervised Head are combined in the final objective function that we use to train our model:

$$\mathcal{L} = \lambda_{ins}\mathcal{L}_{ins} + \lambda_{clu}\mathcal{L}_{clu} + \lambda_{human}\mathcal{L}_{human} \tag{4}$$

where $\lambda_{ins} > 0, \lambda_{clu} > 0$, and $\lambda_{human} > 0$ are hyperparameters.

## 4    Experimental Setup

We use the Stretch RE-1 robot [38] with a static push plate mounted on a telescopic arm to extract items from a moving conveyor belt (as shown in the supplementary video). The robot can extract items by either extending its telescopic arm to push objects off the conveyor belt or pulling objects towards itself. As shown in Fig. 2, the setup consists of a 60-gallon hopper, a small feeder belt, and the main conveyor belt (10 feet long and 1.5 feet wide). We position three cameras above the belt to monitor the recycling stream.

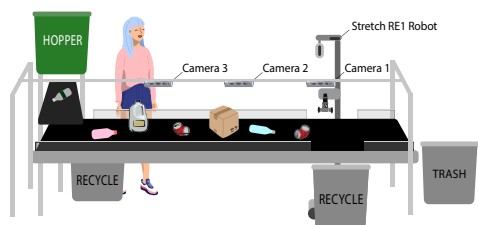

Figure 2: **Experimental Setup.** The human and the robot work together on the recycling conveyor belt to sort the recyclables.

We use over 500 unique recyclable items of various sizes, shapes, colors, and forms in our experiments. The recyclables belong to ten categories: *crushed metal cans, un-crushed metal cans, crushed plastic bottles, un-crushed plastic bottles, un-crushed colored plastic bottles, brown cardboard boxes, coated cardboard boxes, cardboard trays, half-gallon milk jugs*, and *one-gallon milk jugs*. For randomizing the stream, the hopper dispenses a random selection of recyclables onto the feeder belt, which then transports the items to the main conveyor belt. A human can pick up some items of a given category by standing next to the conveyor belt. The robot is expected to sort items similar to the human-selected objects and extract them from the stream by pushing or pulling them off the conveyor belt.

## 5    Offline Evaluation

**Data Collection:** For evaluating the system offline, we created a dataset by having a selection of recyclables pass through the setup described in Section 4. We used YOLACT [39], a real-time instance segmentation model with a ResNet-101 [40] backbone, to predict a rotated bounding box over each item on the conveyor belt. We trained this instance segmentation model with 411 conveyor-belt images containing 4419 instances of recyclables, which were manually annotated with an instance mask. Using an 80% train, 20% validation split, we obtained an F1-score of 0.89 after training the YOLACT instance segmentation model for 100 epochs using the MM-Detection [41] library in PyTorch. Each predicted mask was converted to a rotated bounding box whose crops were extracted for evaluation. Finally, we created a dataset containing 1502 variable-sized crops of recyclables, where each crop belongs to one of the ten categories described in Section 4.

**Evaluation Protocol:** For training, we select three random sets of 40 examples from each of our 10 categories to serve as the human-selected pool for a given category (resulting in 30 human pools in total). We use each human-selected pool to train our proposed model. During inference, we compute the pairwise cosine similarity of every object not selected by the human with every object in a given human-selected pool. We then compute the average cosine similarity of the top-5 most similar objects and assign it as the similarity score of the candidate object. We compute an F1-score per category by thresholding each similarity score between [0.1, 0.95] and assigning a candidate object to a given category if the similarity score is above the threshold [42, 43]. We report F1-scores for the best threshold averaged over three human-selected pools per category.

**Quantitative Results:** Experiment 7 in Fig. 3 shows the performance of our method on the ten categories specified in Section 4 using the evaluation protocol explained above. We compared our method with a supervised learning method comprised of only our human supervised head in isolation. This head effectively learns to classify instances when human-selected data from only one positive class is available during training [44]. Experiment 1 in Fig. 3 shows the F1-score of this supervised learning method to be significantly lower than our method.

**Ablation Study:** We conduct an ablation study of our method to evaluate how each component of the loss in eq. (4) contributes to the overall performance. Experiment 1 in Fig. 3 shows the F1-scores of only the human-supervised head across all categories. The human-supervised head performs poorly on its own likely because the data is heavily imbalanced between human-selected and non-human-selected objects. Experiment 2 in Fig. 3 shows the F1-scores of the instance projection head, which on its own is identical to SimCLR [10]. The high F1-scores across three human pools per category show that this head learns a robust visual representation for all categories. Experiment 3 in Fig. 3 shows that the cluster projection head performs poorer than the instance projection head.

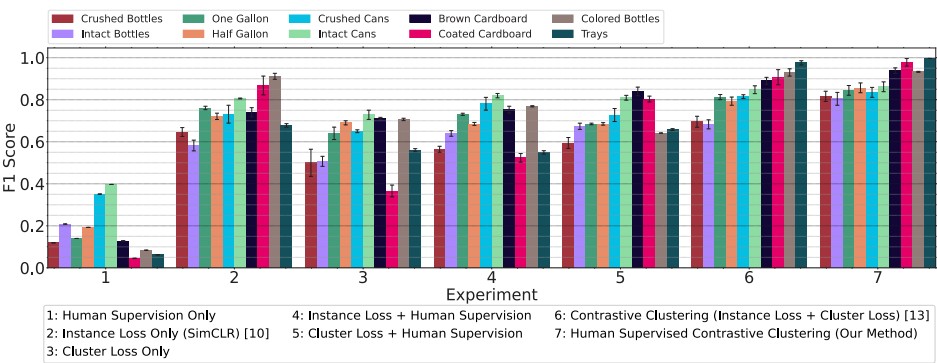

Figure 3: **Results of our Offline Evaluation:** Comparison with Baselines and Ablation Study aggregated over 3 human-selected pools per category

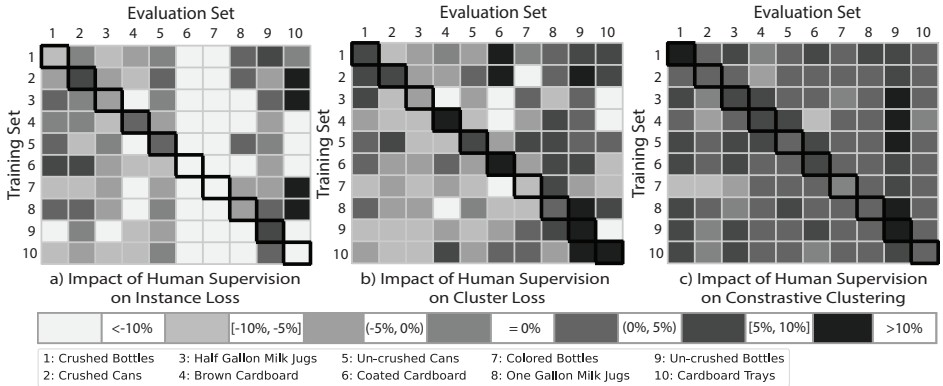

Figure 4: **Impact of augmenting human supervision to contrastive learning**: Difference in performance between self-supervised learning augmented with human supervision and the self-supervised learning model trained independently for instance loss, cluster loss, and contrastive clustering, respectively across categories. Specifically, w.r.t. the experiment numbers in Figure 3, a) represents Experiment 4 - Experiment 2, b) represents Experiment 5 - Experiment 3, c) represents Experiment 7 - Experiment 6.

To assess the impact of human supervision on each of our loss components, we jointly trained each of our self-supervised loss components (instance loss, cluster loss, and contrastive clustering) with our proposed loss that adds human supervision (eq. 3). For these experiments, we trained and evaluated models across three human pools per category (resulting in 30 human pools total). As explained in Section 5, each human pool consists of 40 randomly selected images belonging to a given category. Fig. 4a), 4b) and 4c) summarize the results. Each column in these figures denotes the human pool the method was trained with, and each row denotes the human pool the method was evaluated on. Each cell corresponds to the difference in F1-scores between the respective self-supervised method trained with human supervision and the self-supervised learning method independently.

Fig. 4a) and Experiment 4 in Fig. 3 show that the human-supervised head (eq. 3) does not improve the performance of the instance projection head across all categories. We attribute this result to conflicts between the two objectives. The instance loss maximizes the agreement between two augmentations of the same image while optimizing for pushing other instances away. In contrast, the human supervision head optimizes for pulling human-provided instances together. Fig. 4b) and Experiment 5 in Fig. 3 show that human supervision helps improve the performance of the model trained with the cluster loss for most categories. In this case, the objective functions align with human supervision supporting the formation of meaningful clusters. It is worth noting that when the human pools used for training and evaluation are the same (diagonal elements in Fig. 4b)), human supervision often helps improve the performance of cluster loss.

Finally, we show that our method slightly outperforms the self-supervised contrastive clustering method [13] (Experiment 6 in Fig. 3), showing an average improvement of 4.93% over all cate-

gories. Additionally, Fig. 4c) shows the performance of our method compared to contrastive clustering, which combines the instance and cluster loss. Our method performs well across categories, even when the human pools for training and evaluation belong to different categories (off-diagonal elements in Fig. 4c)). These results show that combining the losses that maximize agreement between both different views of the same instance (instance loss) and different instances belonging to the same cluster (cluster loss) benefit from even limited human supervision. Our human supervision loss facilitates learning a representation with more homogeneous clusters, as seen in the t-SNE plot in the supplementary material.

# 6 A Case Study with a Recycling Robot

This section describes in more detail the system we used to demonstrate our approach in a recyclable sorting task, as shown in Fig. 5. The robot learns the characteristics of objects that a human is sorting, working alongside this person in real-time. Afterward, we detail our experimental design, evaluation protocol, and results.

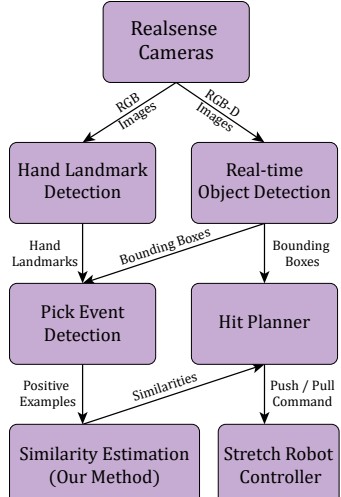

Figure 5: **System Architecture**

## 6.1 System Architecture

**1. Object Detection:** We predict instance segmentation masks over each object on the conveyor belt using the overhead cameras' video streams as described in Section 5. We convert these masks to tightly-fitted rotated bounding boxes.

**2. Detection of Picked Objects:** This module determines which objects the human is picking in real-time. This is done by identifying hand movements associated with an object being picked and then obtaining an un-occluded image of the object from previous frames in the video. The system uses the MediaPipe real-time hand tracking API and a ResNet-18 [40] image classifier to identify between three events: a picking event, no picking, and no hand present over the conveyor belt. To determine which item the human is picking in real-time, the model compares the placement of the person's hand with the position of objects on the belt in a few prior frames.

**3. Similarity Estimation (Our Method):** Our approach allows the robot to learn the features of the objects picked by the person and to remove similar objects out of the items remaining on the conveyor belt. Our human-supervised contrastive clustering model is trained in real-time for 5 epochs on all the objects on the conveyor belt and the human-selected objects by optimizing for eq. 4. The robot then uses the trained learner to identify objects similar to the objects selected by the human. We run inference on every remaining object on the conveyor belt that was not picked by the human and calculate the pairwise cosine similarity between a given object and every object in the human-selected pool. We assign the average pairwise cosine similarity with the positive dataset to be the similarity score of the given object.

**4. Hit Planner:** A hit score is calculated for each object on the belt based on the similarity score of that object and the number of other objects that would likely be removed unintentionally should the robot try to remove the desired object from the conveyor belt. These so-called "casualties" are a consequence of our robot's hardware only being able to push and pull objects off the belt rather than performing more complicated manipulation actions. The object with the highest hit score is targeted by the robot and removed from the belt.

## 6.2 Experimental Protocol and Results

One of the researchers removed recyclables of a given category from the moving conveyor belt. The robot system recorded the items that the human picked from an overhead camera. Once a certain number of items were removed from the stream, the robot trained on the data it had collected. The robot then began to remove items it deemed were similar to human-selected objects from the belt. The human and robot worked together on the same belt, and the robot continued to update its model as it watched the human pick more objects. When the human picked 30 objects from the recycling stream, the robot's model was cleared, and the researcher trained the robot's model for a different category of objects.

We perform inference on images of all objects on the conveyor belt (including the objects selected by the human using our trained model) to obtain their representation in the embedding space. We then compute the pairwise cosine similarity between each human-selected object and every remaining object on the conveyor belt, which are 'candidate objects.' For each candidate object, we average the cosine similarities between itself and the top-5 most similar human-selected objects and assign the average as the similarity score of that candidate object. These similarity scores for the candidate objects on the conveyor belt are leveraged by the hit-planner module to decide what objects are possible to remove with minimal casualties. We manually labeled the video recordings of the robot removing items of a given category to count the number of items correctly/incorrectly removed or missed to calculate the F1-score of the robot.

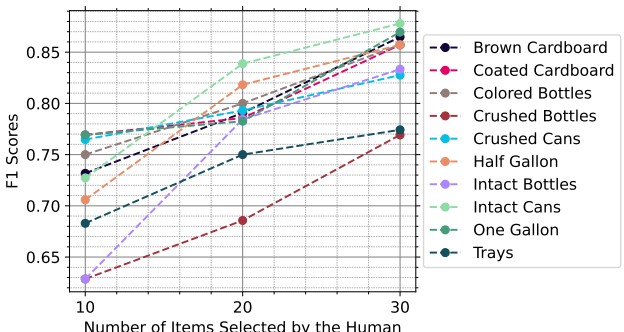

Figure 6: **Model performance in real-time.**

For sample efficiency in real-time, we initialize the robot's model with a pre-trained contrastive clustering model trained on our offline dataset (described in Section 5) with no human supervision. Fig. 6 shows the F1-score of the robot system across all ten categories after the researcher picks 30 items in increments of 10. On average, there were $90 \pm 16$ items on the belt for each round of evaluation (when the human picked ten items). For every 10 human-provided examples, the robot's F1-score improved by 6 points on average across categories.

## 7 Limitations and Future Work

Our evaluation assumes that the human selects objects that belong to the same category. Future experiments will include inter-category object selection and a wider participant pool outside the research team. Hardware constraints also impacted our evaluation. The robot's limited range of motion combined with its relatively slow speed resulted in missing target items and accidental removals. Also, our system assumed the belt's velocity was the velocity of every object. When this assumption breaks, the robot miscalculates objects' distances. These hardware constraints accounted for 36% of all missed items and 32% of all accidental removals across all rounds of evaluation. In the future, the system would benefit from using object tracking algorithms to help the robot identify the movement patterns of each object to make object removal more accurate. Additionally, we evaluated our approach on a relatively small diversity of objects compared to what is expected in a Materials Recovery Facility (MRF) on a relatively sparse and slow belt. Future work should evaluate the performance on a more diverse stream of objects on a fast-moving belt.

Aggregating human feedback at scale is an interesting extension of our work. For example, we imagine a situation where humans observe objects near a robot through a camera feed and provide supervision for which objects to pick remotely. While we don't think that this approach currently makes sense for the recycling domain given economic constraints, future work could gather labels in this manner to accelerate robot learning with our approach. Finally, even though our work focuses on recycling, our approach could also be applied in industrial quality control or the fruit sorting industry. Broadly speaking, our approach could potentially benefit applications where humans and robots work jointly and systematically to manipulate objects. However, since we did not run explicit tests on any of these application domains, we tried not to speculate too broadly in the paper.

## 8 Conclusion

We proposed a novel approach to incorporate human supervision into a contrastive clustering model to better align a robot's visual object representation with humans' requirements. Our method demonstrated improved performance over a supervised learning method and self-supervised learning techniques in a simplified recycling setup. In our case study, we had a human and a robot work alongside each other to sort out recyclables, with the robot learning from the human. We showed that the robot learns representations well suited to the human's requirements with few examples.

**Acknowledgments**

This work was funded by the National Science Foundation (NSF) under grants No. 1955653, 1928448, 2106690, and 1813651. We would like to thank Jake Brawer, Shasvat Desai, Deep Chakraborty, Nicole Salomons, and Nathan Tsoi, who helped improve the paper through their feedback. We would also like to thank Cameron Adams and David Stanley for their help with the supplemental video.

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
