# OpenReview forum: "Tailoring Visual Object Representations to Human Requirements: A Case Study with a Recycling Robot"
_robot-learning.org/CoRL/2022/Conference — CoRL 2022 Poster_

### Official Review · Reviewer_nRzh · 2022-07-30

**Originality:** Fair
**Technical Quality:** Very Good
**Clarity Of Presentation:** Very Good
**Impact:** 3

**Recommendation:**

Weak Accept: I recommend accepting the paper, but will not argue for my recommendation if the majority of other reviewers have a different opinion.

**Summary:**

This paper adapts modern visual object representations based on contrastive losses to better align with human visual categories. At a technical level, the problem is posed as a joint optimization of three objectives that maximize the agreement between: 1) multiple augmented views of the same image, 2) multiple instances from the same category of objects and 3) human-selected examples from a single category. This means that the contrastive component learns a latent space based purely on the visual features of the examples, while the human supervision loss is meant to align that latent space with more “human-like” features. Simulated experiments with perfect “human” input show that adding the supervised loss component does help the contrastive losses recover a representation that better classifies 10 object categories. A conveyor belt scenario with the same objects shows that (with pre-training) the method can correctly classify each category after 30 (expert) examples (per category).

**Issues:**

Please add more discussion addressing the comments above (architecture choices and loss optimization, broad applications beyond object categorization and robotics, feasibility of assumptions).


**Quality Of The Limitations Section:**

Limitations are addressed clearly

**Reviewer Expertise:**

4: The reviewer is confident but not absolutely certain that the evaluation is correct

**Robotics Focus:**

Sufficient demonstration on hardware

**Strengths And Weaknesses:**

I thought the paper was well-written and clear, as was the supplementary video (which I really appreciated as an overview explanation of the paper before digging into the details). The paper is overall well-motivated in the introduction, and the experimental results and ablation well investigated. I also liked the fact that the experiments have object categories that are more interesting than the typical computer vision ones (crushed cans or bottles, intact cans or bottles, etc.). Lastly, I appreciated having a real robot demo on one of the more interesting applications I’ve seen in robotics recently. Despite these strengths, however, I still do have some issues with the paper that I’d like to detail.

Perhaps my biggest gripe with the paper is that the technical novelty seems relatively limited: add a supervised loss to two existing losses. The experimental results do help make up for it, in that they demonstrate the usefulness of all 3 losses together in the ablation, but I would’ve appreciated more discussion and insight to make up for the rather simple contribution: 1. What led to choosing the architecture from Fig. 1? Instead of sharing the weights have you tried training the network iteratively with each loss? What’s your intuition behind why the current architecture would work best? How do you tune the lambda terms in Eq. (4)? 2. Does this method apply more broadly to more than just object properties or robotics? It seems like the proposed pipeline could work for more than just object classification or the conveyor belt example; perhaps a discussion expanding on this was left out to artificially fit into the CoRL robotics theme more strongly, but I would appreciate having it (maybe in the Conclusion). 3. In what settings would the assumption that the human always selects objects from the same category would apply? The conveyor belt application does seem a bit contrived with this assumption in place (I’d normally pick objects from any category and just place them in different bins, rather than sequentially clear them out one category after the other).

**Summary Of Recommendation:**

Overall, the technical novelty and originality of the work is weak, but the motivation is compelling and (I believe) well-timed, the results are compelling, and the application is interesting. As a result, I am leaning weak accept, but I look forward to more discussion as my opinion can be swayed in either direction.

---

> ### Author Response · Authors · 2022-08-24
> **Response to Reviewer nRzh (Part 1/3)**
>
> Thank you for your detailed and thoughtful comments about our work. We are encouraged by the remarks that “the experiments have object categories that are more interesting than the typical computer vision ones” and “ real robot demo [being] one of the more interesting applications I’ve seen in robotics recently”. We address each concern about our paper individually:
>
> `In what settings would the assumption that the human always selects objects from the same category would apply? The conveyor belt application does seem a bit contrived with this assumption in place (I’d normally pick objects from any category and just place them in different bins, rather than sequentially clear them out one category after the other).`
>
> The assumption that a human would manipulate objects from a single object category holds across many real-world Materials Recovery Facilities (MRF), which our setup models. When we started this project, we visited a recycling facility to understand how each category of recyclables is sorted sequentially using a combination of automated sorters and humans. At every stage in the sorting process, a particular category of recyclable was extracted from the stream using an automatic sorter. The items of a given category missed by the automatic sorter were sorted by human sorters who tried manually extracting the remaining objects on a fast-moving and crowded conveyor belt. The human sorters are trained to extract only one category of interest at a given time because some objects in the recycling stream are more valuable than others. For example, after an automatic sorter has extracted most of the paper from the stream, the human sorters would be tasked with extracting all the remaining paper-based products. This would make the stream more uniform for subsequent automatic sorters to remove a certain kind of plastic. In practice, though, the human sorters may not be able to extract all the objects of the desired category, which could lead to quality issues in the target end-product. Therefore, if a robot were to assist a human sorter in sorting through a particular kind of object, the robot would have to learn the properties of the objects that its human collaborator is interested in. Our work proposes a solution to this problem whereby the robot observes the human picking up a limited number of objects from a single category and quickly learns to distinguish these types of objects.
>
> `Does this method apply more broadly to more than just object properties or robotics? It seems like the proposed pipeline could work for more than just object classification or the conveyor belt example; perhaps a discussion expanding on this was left out to artificially fit into the CoRL robotics theme more strongly, but I would appreciate having it (maybe in the Conclusion). `
>
> Our work focuses on recycling, but our approach could also be applied in industrial quality control or the fruit sorting industry. Broadly speaking, our approach could potentially benefit applications where humans and robots work jointly and systematically to manipulate objects. However, since we did not run explicit tests on any of these application domains, we tried not to speculate too broadly in the paper. We will add a discussion about these other application domains to the paper's conclusion.

---

> > ### Author Response · Authors · 2022-08-24
> > **Response to Reviewer nRzh (Part 2/3)**
> >
> > `What led to choosing the architecture from Fig. 1?`
> > `What’s your intuition behind why the current architecture would work best?`
> >
> > We wanted to design a system that enables a human to teach a robot partner the kinds of objects that they are sorting with a few examples and that could be demonstrated in the context of recycling in Materials Recovery Facilities (MRFs). In such facilities, the human workers typically sort out one type of recyclables while adapting to variations in task specifications and seasonal changes in the stream composition. Because of how objects are handled in a MRF and the economic reality of the recycling industry, one of our primary problem constraints was that the robot could quickly learn from a limited number of examples of an object category provided by a human.
> > The next paragraphs explain each of our design considerations in detail:
> >
> > 1. **Large object variety** →  In the recycling setting, the objects are not guaranteed to look constant through time because of the way they are handled, their deformable properties, and the frequent introduction of new objects. For a standard supervised learning method to perform well at classifying these objects, it would require training models with large datasets to keep up with the performance requirements. Therefore, we chose to approach this problem from a self-supervised contrastive learning perspective, which could adapt to the changing stream composition by constantly re-training in a more active learning fashion with just a few new human labels. We explored various state-of-the-art contrastive learning techniques for our work and found that these techniques typically pass two stochastically augmented views of the same image through an image classification network and a multi-layer perceptron (MLP) to bring them close to each other in the embedding space. These methods have been shown to learn very strong representations for initializing large neural networks that perform a wide variety of downstream supervised learning tasks like image classification, semantic segmentation, and object detection[1]. However, since we wanted to learn a representation of objects such that similar objects were grouped in the embedding space, we needed to take into account inter-instance similarity. We found contrastive clustering[13] to be the best self-supervised method for learning object representations.
> >
> > 2. **Humans are unable to select all examples of a given category** → In a real-world recycling setting, where the conveyor belts are typically very crowded, the human sorters cannot pick out all the objects that belong to a given category. So, there could be objects left on the conveyor belt that belong to the category of interest. This means that the robot would have to gain an understanding of the properties of the objects of interest to their human partner with a limited number of positive examples but no negative examples. Therefore, we designed the proposed human-supervised loss to force the features of the human-selected examples to be close to each other in the embedding space. When the human-supervised loss is jointly trained with the contrastive clustering objective function, the human supervised loss guides the formation of clusters in the embedding space to be better aligned to the human choices.
> >
> > 3. **Real-time training considerations** → Theoretically, any image classification network could have been used as a backbone for the contrastive learner. However, since we intended to use this model in real-time, we opted for ResNet-18[31], which is lightweight, and has been shown in the literature to be powerful enough to learn good visual representations[10,11].
> >
> > Jointly training the instance loss, cluster loss, and human-supervised loss works well for our application. As the human selects more objects, the representations learned by the network quickly change to reflect the human’s preferences.

---

> > > ### Author Response · Authors · 2022-08-24
> > > **Response to Reviewer nRzh (Part 3/3)**
> > >
> > > `Instead of sharing the weights have you tried training the network iteratively with each loss?`
> > >
> > > Weight sharing across different neural network modules has been shown to be a powerful method for learning strong visual representations [10, 11]. Nevertheless, we tried training the network with each of our loss functions sequentially without weight sharing, per the recommendation in the review. We first trained the network with the instance loss, followed by the cluster loss, and finally, with the human-supervised loss. [This figure](https://drive.google.com/file/d/17WsH8KyZmTkBJ7BNPOCzs3i17vID9wPJ/view?usp=sharing) shows our findings. Weight sharing improves the F1 score of our method across all categories.
> > >
> > > `How do you tune the lambda terms in Eq. (4)?`
> > >
> > > We obtained the best lambda values in Eq. (4) by performing a standard grid search of hyperparameters in the range [0.1 to 1] for each lambda. We found the model to perform best when each loss component is weighted equally. Future work could use more complex techniques, like Bayesian optimization, to tune the lambda values using a more informed search procedure.

---

> > > > ### Comment · Reviewer_nRzh · 2022-08-26
> > > > **Response and thank you**
> > > >
> > > > I thank the authors for their detailed response and discussion! I think including some of this discussion and clarification will be helpful to the paper. I still like the paper and I think my Weak Accept score is appropriate.

---

### Official Review · Reviewer_2cDu · 2022-07-31

**Originality:** Fair
**Technical Quality:** Excellent
**Clarity Of Presentation:** Very Good
**Impact:** 3

**Recommendation:**

Weak Accept: I recommend accepting the paper, but will not argue for my recommendation if the majority of other reviewers have a different opinion.

**Summary:**

The authors introduce a method for training a visual classifier in a few-shot manner from human preferences, for use in real-time robotics scenarios. In particular, they build upon existing methods from contrastive learning (e.g. contrastive clustering), but introduce an additional loss that tries to make the embeddings of all human-picked objects similar. They show through both offline and online experiments that their method shows small, but statistically significant, improvement over the baselines.

**Issues:**

After reading the paper, two clarity issues I noticed are below, and it would be great if these could be addressed.

1. The idea of predefined, fixed-count "clusters" is not adequately introduced, I believe this is due to the authors' familiarity with the related works such that this idea comes naturally. However, as a robotics reviewer not extremely familiar with contrastive learning it was difficult to understand from this paper alone where the clusters came from, and their limitations.

2. While the method for training the embeddings is described in detail, how the embeddings are used to classify an object for picking or not is not immediately obvious. While this similarity score idea is mentioned in the experiments it is still not fully clear what exactly is used for the pick/dont pick decision, and I think this is better suited for the method section.

Additionally, it is a valuable contribution (although maybe too costly at this point) to actually provide the evaluations of scenarios where the human prefers some objects in some clusters but not all.

**Quality Of The Limitations Section:**

Limitations are addressed clearly

**Reviewer Expertise:**

3: The reviewer is fairly confident that the evaluation is correct

**Robotics Focus:**

Sufficient demonstration on hardware

**Strengths And Weaknesses:**

The experiments are a clear strength here: the authors have successfully applied their method for the proposed task of object classification on a conveyor belt, where they set up an experiment where they performed 5 epochs of real-time training on a human subject's preferences for which objects to pick. This experiment validates empirically their claims over the usefulness of their model in their proposed use case, and the paper deserves recognition for this non-toy experiment alone.

The major weaknesses of this paper are indeed the limitations discussed in the limitations section. There is still a reliance on this idea of predefined, fixed-count clusters (which by the way is not explained in detail, see issues section). The human preference is only used as an additional "cluster". The main problem with this kind of approach is that it is not clear what will happen when the human chooses some objects from within a cluster but not others, e.g. the human cluster does not fully intersect or exclude any of the existing clusters. This use case is relevant in that it is not hard to imagine there would be other criteria besides the cluster assignment for how the human picks these objects (otherwise we could just classify based on the cluster label alone), and it is likely that in these cases the similarity score-based approach for classification (which is also not described in detail, see issues section) will lead to lower classification performance. Experiments of this nature were unfortunately not performed, so it is difficult to estimate what the actual impact would be.

It is also a clear weakness of this paper that the introduced method is equivalent to simply introducing an additional, non-exclusive cluster to the existing idea of contrastive clustering, which is mostly a very incremental improvement. However, as an evaluation of this approach to a novel problem (few-shot human in the loop learning), the paper is valuable as a case study.

**Summary Of Recommendation:**

While the proposed method is mostly an incremental improvement over an existing, known approach; it is also an interesting case study of how this approach can be applied to a new & interesting problem in the context of human-in-the-loop learning. As a result, I lean that the paper be accepted, but do have concerns about the limitations of the approach and how these were not exactly addressed (e.g. mostly avoided) in the experiments.

---

> ### Author Response · Authors · 2022-08-23
> **Response to Reviewer 2cDu (Part 1/3)**
>
> We appreciate the constructive feedback provided in the review. Also, we are glad to hear that our paper deserves recognition for our non-toy experiment. Below we address the concerns mentioned in the review:
>
> `The major weaknesses of this paper are indeed the limitations discussed in the limitations section. There is still a reliance on this idea of predefined, fixed-count clusters (which by the way is not explained in detail, see issues section). The human preference is only used as an additional "cluster". The main problem with this kind of approach is that it is not clear what will happen when the human chooses some objects from within a cluster but not others, e.g. the human cluster does not fully intersect or exclude any of the existing clusters. This use case is relevant in that it is not hard to imagine there would be other criteria besides the cluster assignment for how the human picks these objects (otherwise we could just classify based on the cluster label alone), and it is likely that in these cases the similarity score-based approach for classification (which is also not described in detail, see issues section) will lead to lower classification performance. Experiments of this nature were unfortunately not performed, so it is difficult to estimate what the actual impact would be.
> It is also a clear weakness of this paper that the introduced method is equivalent to simply introducing an additional, non-exclusive cluster to the existing idea of contrastive clustering, which is mostly a very incremental improvement. However, as an evaluation of this approach to a novel problem (few-shot human in the loop learning), the paper is valuable as a case study.`
>
> Our work demonstrates how a robot can quickly learn visual object representations that align with a human’s preference and how our proposed approach can be used in real-time in the challenging domain of sorting recyclables. The novelty of this approach does not stem from simply adding a loss to a representation learning algorithm, but doing so in a way that is compatible with robot learning from co-located human collaborators. Close to our work, Wilber et al. (2015) introduced an algorithm called SNaCK that proposes adding a loss function to a visual representation learning process that reflects a human’s preferences. However, SNaCK requires a human to provide a large number of examples of both objects that the human thinks are similar and dissimilar, which would be infeasible to use in real human-robot interaction. On the other hand, our method requires the human to just provide a few example objects of the category they are interested in. These examples can then be used by the robot to learn a strong object representation that aligns well with human preferences.
>
> It is important to note that inductive bias is key for a machine learning model to learn from just a few samples. In our paper, this bias is provided through the cluster loss, which partitions the embedding space into a pre-specified number of clusters that we hope represent object categories – though this is not enforced by our approach. Adding the human-supervised loss to the representation learning process helps guide the formation of the clusters according to the human’s requirements. Worth noting, that knowing the exact number of object categories that the robot needs to handle in practice is not a requirement of our approach. Rather, a ballpark estimate suffices. See our response to o67U for results from a new experiment that shows that our approach is robust to various choices of the number of clusters K used for the cluster loss.
>
> **Reference**
>
> [Wilber et al. (2015)] Wilber, Michael; Kwak, Iljung; Kriegman, David; Belongie, Serge (2015) Learning Concept Embeddings with Combined Human-Machine Expertise International Conference on Computer Vision (ICCV), 2015.

---

> > ### Author Response · Authors · 2022-08-24
> > **Response to Reviewer 2cDu (Part 2/3)**
> >
> > `The idea of predefined, fixed-count "clusters" is not adequately introduced, I believe this is due to the authors' familiarity with the related works such that this idea comes naturally. However, as a robotics reviewer not extremely familiar with contrastive learning it was difficult to understand from this paper alone where the clusters came from, and their limitations.`
> >
> > We base our work on Contrastive Clustering[13], a self-supervised learning method that tries to partition the embedding space into a pre-specified number of clusters based on the inter-instance similarity between object features. Since contrastive clustering is a self-supervised method, it can form clusters that are non-homogenous and overlapping because there is no external supervisory signal guiding the network to distinguish between object instances based on specific object features. Our method attempts to guide the representation learning process by providing a supervisory signal to help inform the formation of clusters according to the human’s requirements. We will clarify this point in the Methods section of the paper to better convey the contributions of our work. We will also ensure this idea is explained clearly in the discussion of our results.
> >
> > `While the method for training the embeddings is described in detail, how the embeddings are used to classify an object for picking or not is not immediately obvious. While this similarity score idea is mentioned in the experiments it is still not fully clear what exactly is used for the pick/dont pick decision, and I think this is better suited for the method section.`
> >
> > We apologize that our evaluation protocol was difficult to understand from the presentation of our paper. We will improve the Evaluation section of our paper with the following explanation:
> >
> > We run inference on images of all objects on the conveyor belt (including the objects selected by the human using our trained model) to obtain their representation in the embedding space. We then compute the pairwise cosine similarity between each human-selected object and every remaining object on the conveyor belt, which are ‘candidate objects.’ For each candidate object, we average the cosine similarities between itself and the top-5 most similar human-selected objects and assign the average as the similarity score of that candidate object. We use these similarity scores to evaluate our model slightly differently in our offline and robot experiments:
> >
> > -  **Offline Experiments:** We assume access to object categories to evaluate our method in our offline experiments. We compute the F1-score per category by thresholding each similarity score between [0.1, 0.95] and assigning a candidate object to a given category if the similarity score is above the given threshold, following the procedure from [33, 34]. We report F1 scores for the best threshold averaged over three human-selected pools per category, as shown in Figure 3.
> >
> > - **Robot Experiments:**  In our robot experiments, the calculated similarity scores for each candidate object on the conveyor belt are leveraged by the hit-planner module to decide what objects are possible to remove with minimal casualties. For more details about the hit-planner module, please refer to Section 1.3 and Figure 4 in the supplementary material. We then evaluate the F1 score of the robot in removing objects of the desired category from the conveyor belt. We perform this evaluation post-hoc by watching the video robot remove objects after it has been trained by the human incrementally (after picking 10, 20, and 30 objects from each of our ten categories). These scores are shown in Figure 6.

---

> > > ### Author Response · Authors · 2022-08-24
> > > **Response to Reviewer 2cDu (Part 3/3)**
> > >
> > > `Additionally, it is a valuable contribution (although maybe too costly at this point) to actually provide the evaluations of scenarios where the human prefers some objects in some clusters but not all.`
> > >
> > > We agree that this will be an interesting experiment that could provide insights into our method. Due to time constraints, we cannot run such an experiment at this time but will discuss this idea in the future work section of the paper. In particular, we hypothesize based on the many experiments that we've conducted thus far that if the fine-grained object category that the human is interested in is visually different from other object categories, then our approach would be able to learn a suitable representation with a small number of examples. For example, imagine a situation where the category of interest is "crushed blue cans" (rather than simply "crushed cans"). If there are other objects in the recycling stream that look like crushed blue cans, our approach would likely confuse them with the human's desired objects. However, if the other objects are visually different, we expect our approach to perform well.

---

### Official Review · Reviewer_o67U · 2022-08-01

**Originality:** Good
**Technical Quality:** Very Good
**Clarity Of Presentation:** Good
**Impact:** 3

**Recommendation:**

Weak Accept: I recommend accepting the paper, but will not argue for my recommendation if the majority of other reviewers have a different opinion.

**Summary:**

The paper proposes a method for learning visual object representations with human supervision. For this purpose, the authors suggest three components: (i) instance prediction for multiple views of the scene, (ii) cluster prediction for varying instances of the same cluster, and (iii) human supervision for human-selected examples. The authors evaluated their method with offline experiments as well as in real-time with a robot in a recycling domain.

**Issues:**

- The paper nicely provides the background on self-supervised and contrastive learning. However, human-in-the-loop robot learning hasn't been mentioned in the related work section, which is one of the main contributions of this paper. It would be nice to include this literature for more comprehensive related work.
- What is $N$ in Equation 1? Is it the number of all positive pairs in a mini-batch?
- The authors mention 80% train and 20% validation set in the data collection. Was there a separate test set for including human supervision?
- Was the model shown in Figure 1 trained end-to-end?
- The offline evaluation results are not easy to follow in the text:
  - As far as I understand, Figure 4 (b) and Experiment 5 in Figure 3 represent the same condition (Cluster Loss + Human Supervision). If it is the case, it would be great to use the same notation in both figures. Also, I am not sure whether Figure 4(c) and experiment 7 in Figure 3 represent the same condition.
  - It is not clear which conditions have been taken from the literature. I think Experiment 2 in Figure 3 is identical to SimCLR [10]; are there any others?
  - The colors in Figure 4 are not straightforward to interpret. It might be better to make this figure gray-scale.

**Quality Of The Limitations Section:**

Limitations are addressed clearly

**Reviewer Expertise:**

3: The reviewer is fairly confident that the evaluation is correct

**Robotics Focus:**

Sufficient demonstration on hardware

**Strengths And Weaknesses:**

Strengths:

The paper focuses on a critical topic and is well motivated. Its technical quality is good, and there are detailed ablation studies showing the contribution of each suggested component. It has also been evaluated in a case study with a recycling robot, and I appreciate the authors for deploying the system to the real world and showing its applicability.

Weaknesses:

The paper is motivated by stating that "a robot may be asked to interact with a wide variety of objects, making it hard or even impossible to pre-program visual object classifiers suitable for the task of interest". However, the suggested method uses a pre-defined number of object categories $C$ and clusters $K$. Consequently, it focuses on a limited number of objects, and handling a wide variety of objects can be challenging for the suggested system in more complex applications.

**Summary Of Recommendation:**

Although the paper focuses on a limited number of objects, and it could have been improved by including more baseline methods from the literature, it presents a strong contribution to a critical topic with good technical quality and detailed analysis.

---

> ### Author Response · Authors · 2022-08-23
> **Response to Reviewer o67U (Part 1/2)**
>
> Thank you for the detailed comments on our paper. We appreciate the recognition of the applicability of our system in the real world. We will incorporate the feedback provided in the review within our paper to significantly improve its presentation.
>
> `The paper is motivated by stating that "a robot may be asked to interact with a wide variety of objects, making it hard or even impossible to pre-program visual object classifiers suitable for the task of interest". However, the suggested method uses a pre-defined number of object categories C and clusters K. Consequently, it focuses on a limited number of objects, and handling a wide variety of objects can be challenging for the suggested system in more complex applications.`
>
> We base our work on a self-supervised clustering algorithm, Contrastive Clustering [13]. Thus, we are required to provide the number of clusters K as a hyperparameter of the model. In the results presented in the paper, we defined the number of clusters K to be the number of categories C to keep our evaluation consistent with the original Contrastive Clustering paper [13]. However, our method does not need to know precisely how many pre-defined object categories are being observed; instead, a ballpark amount suffices. To demonstrate this idea,  we conducted an additional experiment showing that the number of clusters (K) need not be precisely equal to the number of categories (C). As indicated in the table below, if the number of clusters is significantly lower or higher than the number of categories in the downstream task, there is a significant decrease in performance. Otherwise, the performance of our approach is robust to the value of K.
>
> |K  | Average F1 scores across all categories |
> | ---- | -------|
> | 2 |0.820 ± 0.016|
> |8| 0.873 ± 0.02|
> |9|0.882 ± 0.02|
> |**10**|**0.886 ± 0.017**|
> |11|0.883 ± 0.019|
> |12|0.877 ± 0.02|
> |15|0.867 ± 0.021|
> |20|0.847 ± 0.018|
> |50|0.847 ± 0.018|
> |100|0.813 ± 0.024|
> We will add the above analysis to the supplementary material of the paper.
>
> `The paper nicely provides the background on self-supervised and contrastive learning. However, human-in-the-loop robot learning hasn't been mentioned in the related work section, which is one of the main contributions of this paper. It would be nice to include this literature for more comprehensive related work.`
>
> This is an excellent suggestion. In the Related Work section, we will include the following survey about the relevant state-of-the-art techniques in human-in-the-loop robot learning and discuss how this work fits into the broader area (Same as in response to reviewer V8E7).
>
> > There are many ways in which a robot can learn from a human.  The three most common types of demonstrations used for robot learning are kinesthetic teaching, teleoperation, and passive observation [Ravichandar et. al., (2020)]. Our work uses passive observations because of the ease at which they allow the human to teach the robot and the hardware-agnostic nature of this demonstration type. Much of the recent work in the field of learning from passive observations uses reinforcement learning to train a robot. For example, the work of Mukherjee et al. (2022) uses a learned goal proximity function as a dense reward for policy training, and Karnan et al. (2022) propose to learn navigation policies from a single video demonstration. Our work complements these efforts by proposing a novel approach for learning visual object representations that can be used by a robot for manipulation tasks, like sorting recycling streams.
>
> > Typically, within the area of human preference learning, a robot learns the preferences of its human collaborators to perform a task through various reward shaping techniques [Sadigh et al. (2017), Lee et al. (2021)]. Our work is related to this thread of research because we want a robot to learn the preferences of their human collaborator for a sorting task. However, we want the robot to use passive observation rather than more traditional active querying techniques [Wilde et al. (2020a), Wilde et al. (2020b), van Waveren et al. (2022)] because in the recycling domain humans are already busy picking objects from recycling streams. Preference learning in our work then entails having a robot learn the visual characteristics of the categories of objects that the human collaborator is interested in. We propose to approach this problem from the perspective of self-supervised representation learning and incorporate human supervision in a novel manner to ensure that the learned representation aligns well with the human’s preferences.

---

> > ### Author Response · Authors · 2022-08-23
> > **Response to Reviewer o67U (Part 2/2)**
> >
> > `What is N in Equation 1? Is it the number of all positive pairs in a mini-batch?`
> >
> > Yes, the N in equation 1 is the number of all positive pairs in a mini-batch. We will clarify that in the definition of the Instance Loss in the paper.
> >
> > `The authors mention 80% train and 20% validation set in the data collection. Was there a separate test set for including human supervision?`
> >
> > Yes, there was a separate validation set for reporting the F1-scores of our method. Our model was trained on 80% of the dataset and validated on the remaining 20%. We will update the paper with this information.
> >
> > `Was the model shown in Figure 1 trained end-to-end?`
> >
> > Yes, the model was trained end-to-end for both offline evaluation and online experiments on the robot. More details about model training parameters can be found in Section 3 of the supplementary material.
> >
> > `The offline evaluation results are not easy to follow in the text:`
> >
> > `a) As far as I understand, Figure 4 (b) and Experiment 5 in Figure 3 represent the same condition (Cluster Loss + Human Supervision). If it is the case, it would be great to use the same notation in both figures. Also, I am not sure whether Figure 4(c) and experiment 7 in Figure 3 represent the same condition.`
> >
> > Experiment 7 in Figure 3 and Figure 4(c) convey different results. Experiment 7 in Figure 3 shows the performance of our method (contrastive clustering + human supervision) across all object categories when evaluated on three different sets of human-selected examples. Figure 4(c) shows the impact of augmenting human supervision with contrastive clustering. Each cell of Figure 4(c) shows the difference between the model trained using the self-supervised contrastive clustering method (Experiment 5 in Figure 3) and training and evaluating our method with human pools of different categories. Experiment 3 in Figure 3 and Figure 4(b) are related in a similar fashion. Figure 4(b) shows the impact of adding human supervision to just the clustering loss component. Similarly, Figure 4(a) shows the impact of adding human supervision to just the instance loss component (Experiment 2 in Figure 3). As suggested in the review, we will clarify the correspondence between these Figures by making the notations consistent across them.
> >
> > `b) It is not clear which conditions have been taken from the literature. I think Experiment 2 in Figure 3 is identical to SimCLR [10]; are there any others?`
> >
> > Experiments 2 and 6 have been taken from SimCLR[10] and Contrastive Clustering[13], respectively. We consider them baselines from the literature. We will clarify these details in the Experiments section and in the description of Figure 3.
> >
> > `c)The colors in Figure 4 are not straightforward to interpret. It might be better to make this figure gray-scale.`
> >
> > We will make the colors more accessible, and include a table with the corresponding numerical results in the Supplementary Material. Thanks for the suggestion.
> >
> > **References**
> >
> > [Ravichandar et. al., (2020)]  Ravichandar, H., Polydoros, A. S., Chernova, S., & Billard, A. (2020). Recent advances in robot learning from demonstration. Annual review of control, robotics, and autonomous systems, 3, 297-330.
> >
> > [Mukherjee et al. (2022)]  Mukherjee, D., Gupta, K., Chang, L. H., & Najjaran, H. (2022). A survey of robot learning strategies for human-robot collaboration in industrial settings. Robotics and Computer-Integrated Manufacturing, 73, 102231.
> >
> > [Karnan et al. (2022)]  Karnan, H., Warnell, G., Xiao, X., & Stone, P. (2022, May). Voila: Visual-observation-only imitation learning for autonomous navigation. In 2022 International Conference on Robotics and Automation (ICRA) (pp. 2497-2503). IEEE.
> >
> > [Sadigh et al. (2017)] Sadigh, D., Dragan, A. D., Sastry, S., & Seshia, S. A. (2017). Active preference-based learning of reward functions.
> >
> > [Lee et al. (2021)] Lee, Kimin, Laura Smith, and Pieter Abbeel. "Pebble: Feedback-efficient interactive reinforcement learning via relabeling experience and unsupervised pre-training." arXiv preprint arXiv:2106.05091 (2021).
> >
> > [Wilde et al. (2020a)] Wilde, Nils, et al. "Improving user specifications for robot behavior through active preference learning: Framework and evaluation." The International Journal of Robotics Research 39.6 (2020): 651-667.
> >
> > [Wilde et al. (2020b)] Wilde, Nils, Dana Kulić, and Stephen L. Smith. "Active preference learning using maximum regret." 2020 IEEE/RSJ International Conference on Intelligent Robots and Systems (IROS). IEEE, 2020.
> >
> > [van Waveren et al. (2022)]  van Waveren, Sanne, et al. "Correct Me If I'm Wrong: Using Non-Experts to Repair Reinforcement Learning Policies." Proceedings of the 17th ACM/IEEE International Conference on Human-Robot Interaction. 2022.

---

### Official Review · Reviewer_F5P5 · 2022-08-01

**Originality:** Good
**Technical Quality:** Good
**Clarity Of Presentation:** Good
**Impact:** 4

**Recommendation:**

Weak Accept: I recommend accepting the paper, but will not argue for my recommendation if the majority of other reviewers have a different opinion.

**Summary:**

The work integrates human supervision into representation learning process to compute visual object representations inline with human requirements for interest tasks. The work combines existing contrastive learning method with a loss function that brings the representations of human-selected objects close to each other in the latent space. They evaluate the work on a simplified recycling setup that allows human-robot interaction.

**Issues:**

(1) would like to see if authors considered any approaches to alleviate the need of active human feedback by instead relying on approaches that incorporate scalable human feedback ( example : using intermittent oversight or meta-imitation learning agents). (2) would also like to see experiments with more diverse set of objects

**Quality Of The Limitations Section:**

Additional details required

**Reviewer Expertise:**

3: The reviewer is fairly confident that the evaluation is correct

**Robotics Focus:**

Sufficient demonstration on hardware

**Strengths And Weaknesses:**

strengths : (1) adapts to dynamic human-defined requirements by leveraging human-feedback (2) performs both online and offline evaluations, ablations, and outperforms some self-supervised learning baselines. (3) the case study has immediate relevance to real-world, though would like further explanation on the novelty of the architecture.

weaknesses : (1) requires active human supervision which is expensive  (2) the objects used for the experiments are limited in variations to fully evaluate the performance of the system.

**Summary Of Recommendation:**

I read the paper, and used good judgement in assessing the paper.

---

> ### Author Response · Authors · 2022-08-23
> **Response to Reviewer F5P5**
>
> Thank you for the insightful comments. We are excited to see that the review points out that our case-study has immediate relevance to the real world. Below we address individually each of the concerns in the review.
>
> `would like to see if authors considered any approaches to alleviate the need of active human feedback by instead relying on
> approaches that incorporate scalable human feedback ( example: using intermittent oversight or meta-imitation learning agents). `
>
> We agree that human supervision is generally expensive, which is why our proposed system aims to learn from small amounts of human examples. In particular, we designed our robot system to be deployed in environments where robots would work alongside humans to help them complete repetitive tasks. In recycling, such a system requires that a human teaches what they are sorting to their robot partner because the task requirements change constantly.
>
> In the paper, we will mention aggregating human feedback at scale as an interesting extension of our work. For example, we imagine a situation where humans observe objects near a robot through a camera feed and provide supervision for which objects to pick remotely. While we don’t think that this approach makes sense currently for the recycling domain given economic constraints, future work could gather labels in this manner to accelerate robot learning with our approach.
>
>
> `would also like to see experiments with more diverse set of objects`
>
> It is challenging to develop a dataset that includes the enormous variety of recyclables that a real Materials Recovery Facility (MRF) processes. For our research, we toured such facilities and watched numerous videos documenting the composition of streams at MRFs. Then, we worked to create a collection of objects representing a characteristic sampling of recyclables typically seen in these facilities. More specifically, we selected our set of over 500 unique objects that were diverse in category, material, size, color, shape, deformability, reflectiveness, dirtiness, opacity, and density. We deliberately excluded paper-based and glass-based recyclables because air pumps and optical sorters typically remove these items reasonably well in a MRF.
>
> For our experiments, we captured images of the selected objects under different lighting conditions, orientations, and levels of motion blur on a moving conveyor belt, adding to the diversity of our dataset. We will add [this image](https://drive.google.com/file/d/1h4slxbTLHGL9hBfu2pJzV256JU0G0NY6/view?usp=sharing) that shows some of the variations mentioned above to the supplementary material. Our experimental setup is a necessary first step toward validating our system architecture under the harsh conditions typically encountered in a MRF.

---

### Meta-Review · Area_Chair_V8E7 · 2022-08-12

**Recommendation:** Accept (Poster)
**Confidence:** 5

**Metareview:**

The paper presents a method for human-centric object recognition, with the application domain being the recycling conveyor belts, where human and robot can co-exist.

The paper's **strengths** are:
- the compelling motivation of the proposed method;
- the clear technical description of the work;
- the interesting dataset used in this work;
- the real-world demonstration is very promising.

The **weaknesses** of the paper are:
- while the simple addition of the supervisory seems to work, additional insights regarding the architecture are missing;
- the paper does not discuss the state-of-the-art of human-centric robot learning, and human preference-based learning in their paper;
- some missing information regarding the method's performance given the sparse labeling from the human feedback.

I think the authors can majorly improve the presentation of their work, once they address the raised issues in the rebuttal period.

**Post rebuttal assessment:** The paper introduces a very interesting and timely application domain. The methodological contribution is not as strong as the experimental part, however, the paper introduces an interesting system, and as such can be very beneficial for the community.

**Best Paper Nomination:**

No

---

> ### Author Response · Authors · 2022-08-24
> **Response to Area Chair V8E7 (Part 1/2)**
>
> We would first like to thank the reviewers and the AC for the thoughtful and constructive comments. We appreciate the reviewers’ remarks that:
> - our work focuses on a critical topic and is well motivated (reviewer o67u),
> - we evaluated our method extensively in the recycling domain (reviewers nRzh and 2cDu),
> - our case study has immediate relevance to the real-world (reviewer F5P5) and,
> - our paper deserves recognition for the non-toy experiment (reviewer 2cDu).
> We address all of the reviewers' concerns by replying to each of them individually.
>
> `while the simple addition of the supervisory seems to work, additional insights regarding the architecture are missing;`
>
> We detailed our design decisions in designing the model architecture in the following response to reviewer nRzh, and plan to include those insights in different parts of our paper to clarify our rationale for the proposed approach.
>
>
> > We wanted to design a system that enables a human to teach a robot partner the kinds of objects that they are sorting with a few examples and that could be demonstrated in the context of recycling in Materials Recovery Facilities (MRFs). In such facilities, the human workers typically sort out one type of recyclables while adapting to variations in task specifications and seasonal changes in the stream composition. Because of how objects are handled in a MRF and the economic reality of the recycling industry, one of our primary problem constraints was that the robot could quickly learn from a limited number of examples of an object category provided by a human.
> The next paragraphs explain each of our design considerations in detail:
>
> > 1. **Large object variety** →  In the recycling setting, the objects are not guaranteed to look constant through time because of the way they are handled, their deformable properties, and the frequent introduction of new objects. For a standard supervised learning method to perform well at classifying these objects, it would require training models with large datasets to keep up with the performance requirements. Therefore, we chose to approach this problem from a self-supervised contrastive learning perspective, which could adapt to the changing stream composition by constantly re-training in a more active learning fashion with just a few new human labels.
> We explored various state-of-the-art contrastive learning techniques for our work and found that these techniques typically pass two stochastically augmented views of the same image through an image classification network and a multi-layer perceptron (MLP) to bring them close to each other in the embedding space. These methods have been shown to learn very strong representations for initializing large neural networks that perform a wide variety of downstream supervised learning tasks like image classification, semantic segmentation, and object detection[1]. However, since we wanted to learn a representation of objects such that similar objects were grouped in the embedding space, we needed to take into account inter-instance similarity. We found contrastive clustering[13] to be the best self-supervised method for learning object representations.
>
> > 2. **Humans are unable to select all examples of a given category** → In a real-world recycling setting, where the conveyor belts are typically very crowded, the human sorters cannot pick out all the objects that belong to a given category. So, there could be objects left on the conveyor belt that belong to the category of interest. This means that the robot would have to gain an understanding of the properties of the objects of interest to their human partner with a limited number of positive examples but no negative examples. Therefore, we designed the proposed human-supervised loss to force the features of the human-selected examples to be close to each other in the embedding space. When the human-supervised loss is jointly trained with the contrastive clustering objective function, the human supervised loss guides the formation of clusters in the embedding space to be better aligned to the human choices.
>
> > 3. **Real time training considerations** → Theoretically, any image classification network could have been used as a backbone for the contrastive learner. However, since we intended to use this model in real time, we opted for ResNet-18, which is lightweight, and has been shown in the literature to be powerful enough to learn good visual representations[10,11].
> Jointly training the instance loss, cluster loss and human supervised loss works well for our application. As the human selects more objects, the representations learned by the network quickly change to reflect the human’s preferences.

---

> > ### Author Response · Authors · 2022-08-24
> > **Response to Area Chair V8E7 (Part 2/2)**
> >
> > `the paper does not discuss the state-of-the-art of human-centric robot learning and human preference-based learning in their paper;`
> >
> > We will strengthen the related work section of the paper with the following text, which will  provide an overview of the literature in human-centric learning and help situate the proposed work into this area.
> >
> > > There are many ways in which a robot can learn from a human.  The three most common types of demonstrations used for robot learning are kinesthetic teaching, teleoperation, and passive observation [Ravichandar et. al., (2020)]. Our work uses passive observations because of the ease at which they allow the human to teach the robot and the hardware-agnostic nature of this demonstration type. Much of the recent work in the field of learning from passive observations uses reinforcement learning to train a robot. For example, the work of Mukherjee et al. (2022) uses a learned goal proximity function as a dense reward for policy training, and Karnan et al. (2022) propose to learn navigation policies from a single video demonstration. Our work complements these efforts by proposing a novel approach for learning visual object representations that can be used by a robot for manipulation tasks, like sorting recycling streams.
> >
> > > Typically within the area of human preference learning, a robot learns the preferences of its human collaborators to perform a task through various reward shaping techniques [Sadigh et al. (2017), Lee et al. (2021)]. Our work is related to this thread of research because we want a robot to learn the preferences of their human collaborator for a sorting task. However, we want the robot to use passive observation rather than more traditional active querying techniques [Wilde et al. (2020a), Wilde et al. (2020b), van Waveren et al. (2022)] because in the recycling domain humans are already busy picking objects from recycling streams. Preference learning in our work then entails having a robot learn the visual characteristics of the categories of objects that the human collaborator is interested in. We propose to approach this problem from the perspective of self-supervised representation learning and incorporate human supervision in a novel manner to ensure that the learned representation aligns well with the human’s preferences.
> >
> > > **References**
> >
> > > [Ravichandar et. al., (2020)]  Ravichandar, H., Polydoros, A. S., Chernova, S., & Billard, A. (2020). Recent advances in robot learning from demonstration. Annual review of control, robotics, and autonomous systems, 3, 297-330.
> >
> > > [Mukherjee et al. (2022)]  Mukherjee, D., Gupta, K., Chang, L. H., & Najjaran, H. (2022). A survey of robot learning strategies for human-robot collaboration in industrial settings. Robotics and Computer-Integrated Manufacturing, 73, 102231.
> >
> > > [Karnan et al. (2022)]  Karnan, H., Warnell, G., Xiao, X., & Stone, P. (2022, May). Voila: Visual-observation-only imitation learning for autonomous navigation. In 2022 International Conference on Robotics and Automation (ICRA) (pp. 2497-2503). IEEE.
> >
> > > [Sadigh et al. (2017)] Sadigh, D., Dragan, A. D., Sastry, S., & Seshia, S. A. (2017). Active preference-based learning of reward functions.
> >
> > > [Lee et al. (2021)] Lee, Kimin, Laura Smith, and Pieter Abbeel. "Pebble: Feedback-efficient interactive reinforcement learning via relabeling experience and unsupervised pre-training." arXiv preprint arXiv:2106.05091 (2021).
> >
> > > [Wilde et al. (2020a)] Wilde, Nils, et al. "Improving user specifications for robot behavior through active preference learning: Framework and evaluation." The International Journal of Robotics Research 39.6 (2020): 651-667.
> >
> > > [Wilde et al. (2020b)] Wilde, Nils, Dana Kulić, and Stephen L. Smith. "Active preference learning using maximum regret." 2020 IEEE/RSJ International Conference on Intelligent Robots and Systems (IROS). IEEE, 2020.
> >
> > > [van Waveren et al. (2022)]  van Waveren, Sanne, et al. "Correct Me If I'm Wrong: Using Non-Experts to Repair Reinforcement Learning Policies." Proceedings of the 17th ACM/IEEE International Conference on Human-Robot Interaction. 2022.
> >
> > `some missing information regarding the method's performance given the sparse labeling from the human feedback.`
> >
> > We are eager to address this concern, but we are unsure about what information is missing about the method’s performance. Can you clarify this question?

---

> ### Author Response · Authors · 2022-08-28
> **Revised Manuscript**
>
> **Comment:**
>
> We thank the reviewers for their suggestions that helped improve the paper. The revised manuscript and the supplementary material are attached. The revised text is colored in blue.
>
> **Zip File:**
>
> /attachment/4882bc1c5cc0460b524392d51ab857e2ff5a8a4c.zip